# Entropy Coding of Unordered Data Structures

**Julius Kunze**
University College London
juliuskunze@gmail.com

**Daniel Severo**
University of Toronto and Vector Institute
d.severo@mail.utoronto.ca

**Giulio Zani**
University of Amsterdam
g.zani@uva.nl

**Jan-Willem van de Meent**
University of Amsterdam
j.w.vandemeent@uva.nl

**James Townsend**
University of Amsterdam
j.h.n.townsend@uva.nl

### Abstract

We present shuffle coding, a general method for optimal compression of sequences of unordered objects using bits-back coding. Data structures that can be compressed using shuffle coding include multisets, graphs, hypergraphs, and others. We release an implementation that can easily be adapted to different data types and statistical models, and demonstrate that our implementation achieves state-of-the-art compression rates on a range of graph datasets including molecular data.

## 1 Introduction

The information stored and communicated by computer hardware, in the form of strings of bits and bytes, is inherently ordered. A string has a first and last element, and may be indexed by numbers in $\mathbb{N}$, a totally ordered set. For data like text, audio, or video, this ordering carries meaning. However, there are also numerous data structures in which the 'elements' have no meaningful order. Common examples include graphs, sets and multisets, and 'map-like' datatypes such as JSON. Recent applications of machine learning to molecular data benefit from large datasets of molecules, which are graphs with vertex and edge labels representing atom and bond types (some examples are shown in Table 1 below). All of these data are necessarily stored in an ordered manner on a computer, but the order then represents *redundant information*. This work concerns optimal lossless compression of unordered data, and we seek to eliminate this redundancy.

Table 1: Examples of molecules and their order information. The 'discount' column shows the saving achieved by shuffle coding by removing order information (see eq. 14). For each molecule $\mathbf{m}$, $n$ is the number of atoms and $|\mathrm{Aut}(\mathbf{m})|$ is the size of the automorphism group. All values are in bits, and $\log$ denotes the binary logarithm.

| Molecular structure | | Permutation $\log n!$ | Symmetry $\log|\mathrm{Aut}(\mathbf{m})|$ | Discount $\log n! - \log|\mathrm{Aut}(\mathbf{m})|$ |
|---|---|---|---|---|
| Nitric oxide | N=O | 1.00 | 0.00 | 1.00 |
| Water | H\O\H | 2.58 | 1.00 | 1.58 |
| Hydrogen peroxide | H\O−O\H | 4.58 | 1.00 | 3.58 |
| Ethylene | H₂C=CH₂ | 9.49 | 3.00 | 6.49 |
| Boric acid | H-O B-O O-H | 12.30 | 2.58 | 9.71 |

Recent work by Severo et al. (2023a) showed how to construct an optimal lossless codec for (unordered) multisets from a codec for (ordered) vectors, by storing information in an ordering. Their method depends on the simple structure of multisets' automorphism groups, and does not extend to other unordered objects such as unlabeled graphs. In this paper we overcome this issue and develop *shuffle coding*, a method for constructing codecs for general 'unordered objects' from codecs for 'ordered objects'. Our definitions of ordered and unordered objects are based on the concept of 'combinatorial species' (Bergeron et al., 1997; Joyal, 1981), originally developed to assist with the enumeration of combinatorial structures. They include multisets, as well as all of the other unordered data structures mentioned above, and many more.

Although the method is applicable to any unordered object, we focus our experiments on unordered (usually referred to as 'unlabeled') graphs, as these are a widely used data type, and the improvements in compression rate from removing order information are large (as summarized in Table 2). We show that shuffle coding can achieve significant improvements relative to existing methods, when compressing unordered graphs under the Erdős-Rényi $G(n, p)$ model of Erdős and Rényi (1960) as well as the recently proposed Pólya's urn-based model of Severo et al. (2023b). Shuffle coding extends to graphs with vertex and edge attributes, such as the molecular and social network datasets of TUDatasets (Morris et al., 2020), which are compressed in Section 5. We release source code[1] with straightforward interfaces to enable future applications of shuffle coding with more sophisticated models and to classes of unordered objects other than graphs.

## 2 BACKGROUND

The definitions for ordered and unordered objects are given in Section 2.1. Entropy coding is reviewed in Section 2.2. Examples are given throughout the section for clarification.

### 2.1 PERMUTABLE CLASSES

For $n \in \mathbb{N}$, we let $[n] := \{0, 1, \ldots, n - 1\}$, with $[0] = \emptyset$. The symmetric group of permutations on $[n]$, i.e. bijections from $[n]$ to $[n]$, will be denoted by $\mathcal{S}_n$. Permutations compose on the left, like functions, i.e., for $s, t \in \mathcal{S}_n$, the product $st$ denotes the permutation formed by performing $t$ then $s$.

Permutations are represented as follows

$$\begin{smallmatrix} 0 \leftarrow 1 \\ \searrow \nearrow \\ 2 \end{smallmatrix} = (2, 0, 1) \in \mathcal{S}_3. \tag{1}$$

The glyph on the left-hand side represents the permutation that maps 0 to 2, 1 to 0 and 2 to 1. This permutation can also be represented concretely by the vector $(2, 0, 1)$.

Concepts from group theory, including subgroups, cosets, actions, orbits, and stabilizers are used throughout. We provide a brief introduction in Appendix A.

We will be compressing objects which can be 're-ordered' by applying permutations. This is formalized in the following definition:

**Definition 2.1** (Permutable class[2]). For $n \in \mathbb{N}$, a *permutable class* of order $n$ is a set $\mathcal{F}$, equipped with a left group action of the permutation group $\mathcal{S}_n$ on $\mathcal{F}$, which we denote with the $\cdot$ binary operator. We refer to elements of $\mathcal{F}$ as *ordered objects*.

**Example 2.1** (Length $n$ strings). For a fixed set $X$, let $\mathcal{F}_n = X^n$, that is, length $n$ strings of elements of $X$, and let $\mathcal{S}_n$ act on a string in $\mathcal{F}_n$ by rearranging its elements.

Taking $X$ to be the set of ASCII characters, we can define a permutable class of ASCII strings, with action by rearrangement, for example

$$\begin{smallmatrix} 0 \leftarrow 1 \\ \searrow \uparrow \\ \circlearrowright 3 \quad 2 \end{smallmatrix} \cdot \texttt{"Team"} = \texttt{"eaTm"}. \tag{2}$$

---

[1] Source code, data and results are available at `https://github.com/juliuskunze/shuffle-coding`.

[2] This definition is very close to that of a 'combinatorial species', the main difference being that we fix a specific $n$. See discussion in Yorgey (2014, pp. 66–67).

**Example 2.2** (Simple graphs $\mathcal{G}_n$)**.** Let $\mathcal{G}_n$ be the set of simple graphs with vertex set $[n]$. Specifically, an element $g \in \mathcal{G}_n$ is a set of 'edges', which are unordered pairs of elements of $[n]$. We define the action of $\mathcal{S}_n$ on a graph by moving the endpoints of each edge in the direction of the arrows, for example

$$
\begin{array}{cc}
0 \leftarrow 1 \\
\searrow \uparrow \\
\circlearrowright 3 \quad 2
\end{array}
\cdot
\begin{array}{cc}
0 - 1 \\
\searrow | \\
3 - 2
\end{array}
=
\begin{array}{cc}
0 - 1 \\
\times | \\
3 \quad 2
\end{array}. \tag{3}
$$

Our main contribution in this paper is a general method for compressing *unordered* objects. These may be defined formally in terms of the equivalence classes, known as orbits, which comprise objects that are identical up to re-ordering (see Appendix A for background):

**Definition 2.2** (Isomorphism, unordered objects)**.** For two objects $f$ and $g$ in a permutable class $\mathcal{F}$, we say that $f$ is *isomorphic* to $g$, and write $f \simeq g$, if there exists $s \in \mathcal{S}_n$ such that $g = s \cdot f$ (i.e. if $f$ and $g$ are in the same orbit under the action of $\mathcal{S}_n$). Note that the relation $\simeq$ is an equivalence relation. For $f \in \mathcal{F}$ we use $\tilde{f}$ to denote the equivalence class containing $f$, and $\widetilde{\mathcal{F}}$ to denote the quotient set of equivalence classes. We refer to elements $\tilde{f} \in \widetilde{\mathcal{F}}$ as *unordered objects*.

For the case of strings in example 2.1, an object's isomorphism class is characterized by the multiset of elements contained in the string. For the simple graphs in example 2.2, the generalized isomorphism in definition 2.2 reduces to the usual notion of graph isomorphism. We can define a shorthand notation for unordered graphs, with points at the nodes instead of numbers:

$$
\begin{array}{cc}
\cdot - \cdot \\
\searrow | \\
\cdot - \cdot
\end{array}
:=
\widetilde{
\begin{array}{cc}
0 - 1 \\
\searrow | \\
3 - 2
\end{array}
}. \tag{4}
$$

Using this notation, the unordered simple graphs on three vertices, for example, can be written:

$$
\widetilde{\mathcal{G}_3} = \left\{ \; \begin{array}{c}\cdot \quad \cdot \\ \cdot \end{array} , \; \begin{array}{c}\cdot - \cdot \\ \cdot \end{array} , \; \begin{array}{c}\cdot \\ \searrow \\ \cdot \quad \cdot \end{array} , \; \begin{array}{c}\cdot \\ \diagdown\diagup \\ \cdot \end{array} \right\}. \tag{5}
$$

Finally, we define the subgroup of $\mathcal{S}_n$ which contains the symmetries of a given object $f$:

**Definition 2.3** (Automorphism group)**.** For an element $f$ of a permutable class $\mathcal{F}$, we let $\mathrm{Aut}(f)$ denote the *automorphism group* of $f$, defined by

$$
\mathrm{Aut}(f) := \{ s \in \mathcal{S}_n \mid s \cdot f = f \}. \tag{6}
$$

This is the stabilizer subgroup of $f$ under the action of $\mathcal{S}_n$.

The elements of the automorphism group of the simple graph from example 2.2 are:

$$
\mathrm{Aut}\left( \begin{array}{cc} 0 - 1 \\ \searrow | \\ 3 - 2 \end{array} \right) = \left\{ \begin{array}{cc} \circlearrowright 0 & 1 \circlearrowleft \\ \circlearrowright 3 & 2 \circlearrowleft \end{array} , \quad \begin{array}{cc} 0 \circlearrowright\circlearrowleft 1 \\ \circlearrowright 3 \quad 2 \circlearrowleft \end{array} \right\}. \tag{7}
$$

### 2.1.1 Canonical orderings

To define a codec for unordered objects, we will introduce the notion of a 'canonical' representative of each equivalence class in $\widetilde{\mathcal{F}}$. This allows us, for example, to check whether two ordered objects are isomorphic, by mapping both to the canonical representative and comparing.

**Definition 2.4** (Canonical ordering)**.** A *canonical ordering* is an operator $\bar{\cdot} : \mathcal{F} \to \mathcal{F}$, such that

1. For $f \in \mathcal{F}$, we have $\overline{f} \simeq f$.

2. For $f, g \in \mathcal{F}$, $\overline{f} = \overline{g}$ if and only if $f \simeq g$.

For strings, any sorting function satisfies properties 1 and 2 and is therefore a valid canonical ordering. For graphs, the canonical orderings we use are computed using the `nauty` and `Traces` libraries (McKay and Piperno, 2014). The libraries provide a function, which we call `canon_perm`, which, given a graph $g$, returns a permutation $s$ such that $s \cdot g = \overline{g}$. In addition to `canon_perm`, `nauty` and

`Traces` can compute the automorphism group of a given graph, via a function which we refer to as `aut`.[3]

Our method critically depends on the availability of such a function for a given permutation class. While permutable objects other than graphs cannot be directly canonized by `nauty` and `Traces`, it is often possible to embed objects into graphs in such a way that the structure is preserved and the canonization remains valid (see Anders and Schweitzer (2021)). We use an embedding of edge-colored graphs into vertex-colored graphs in order to canonize and compress graphs with edge attributes (which are not directly supported by nauty/traces). We leave more systematic approaches to canonizing objects from permutable classes as an interesting direction for future work[4].

## 2.2 CODECS

We fix a set $M$ of prefix-free binary messages, and a length function $l: M \to [0, \infty)$, which measures the number of physical bits required to represent values in $M$. Our method requires stack-like (LIFO) codecs, such as those based on the range variant of asymmetric numeral systems (rANS), to save bits corresponding to the redundant order using bits-back (Townsend et al., 2019).

**Definition 2.5** ((Stack-like) codec). A *stack-like codec* (or simply *codec*) for a set $X$ is an invertible function

$$\text{encode} : M \times X \to M. \tag{8}$$

We call a codec *optimal* for a probability distribution over $X$ with mass function $P$ if for any $m \in M$ and $x \in X$,

$$l(\text{encode}(m, x)) \approx l(m) + \log \frac{1}{P(x)}.^5 \tag{9}$$

We refer to $\log \frac{1}{P(x)}$ as the *optimal rate* and to the inverse of encode as decode. Since decode has to be implemented in practice, we treat it as an explicit part of a codec below.

The encode function requires a pre-existing message as its first input. Therefore, at the beginning of encoding we set $m$ equal to some fixed, short initial message $m_0$, with length less than 64 bits. As in other entropy coding methods, which invariably have some small constant overhead, this 'initial bit cost' is amortized as we compress more data.

We will assume access to three primitive codecs provided by rANS. These are

- `Uniform(n)`, optimal for a uniform distribution on $\{0, 1, \dots, n\text{-}1\}$.
- `Bernoulli(p)`, optimal for a Bernoulli distribution with probability `p`.
- `Categorical(ps)`, optimal for a categorical distribution with probability vector `ps`.

These primitive codecs can be composed to implement codecs for strings and simple graphs. In appendix B, we show such a string codec optimal for a distribution where each character is drawn i.i.d. from a categorical with known probabilities, and a codec for simple graphs optimal for the Erdős-Rényi $G(n, p)$ model, where each edge's existence is decided by an independent draw from a Bernoulli with known probability parameter. We will use these codecs for ordered objects as a component of shuffle coding.

There is an implementation-dependent limit on the parameter `n` of `Uniform` and on the number of categories for `Categorical`. In the 64-bit rANS implementation which we wrote for our experiments, this limit is $2^{48}$. This is not large enough to, for example, cover $\mathcal{S}_n$ for large $n$, and therefore permutations must be encoded and decoded sequentially, see Appendix C. For details on the implementation of the primitive rANS codecs listed above, see Duda (2009) and Townsend (2021).

---

[3]In fact, a list of generators for the group is computed, rather than the entire group, which may be very large.

[4]Schweitzer and Wiebking (2019) describe generic methods for canonization starting from a constructive definition of permutable objects using 'hereditarily finite sets' (i.e. not using the species definition).

[5]This condition, with a suitable definition of $\approx$, is equivalent to rate-optimality in the usual Shannon sense, see Townsend (2020).

Figure 1: Visualization of lemma 3.2. For a fixed graph $g$, the six elements $s \in \mathcal{S}_3$ can be partitioned according to the value of $s \cdot g$. The three sets in the partition are the left cosets of $\mathrm{Aut}(g)$.

## 3 CODECS FOR UNORDERED OBJECTS

Our main contribution in this paper is a generic codec for unordered objects, i.e. a codec respecting a given probability distribution on $\widetilde{\mathcal{F}}$. We first derive an expression for the optimal rate that this codec should achieve, then in Section 3.1 we describe the codec itself.

To help simplify the presentation, we will use the following generalization of exchangeability from sequences of random variables to arbitrary permutable classes:

**Definition 3.1** (Exchangeability). For a probability distribution $P$ defined on a permutable class $\mathcal{F}$, we say that $P$ is *exchangeable* if isomorphic objects have equal probability under $P$, i.e. if

$$f \simeq g \Rightarrow P(f) = P(g). \tag{10}$$

We can assume, without loss of modeling power, that unordered objects are generated by first generating an ordered object $f$ from an exchangeable distribution and then 'forgetting' the order by projecting $f$ onto its isomorphism class $\tilde{f}$:

**Lemma 3.1** (Symmetrization). *For any distribution $Q$ on a class of unordered objects $\widetilde{\mathcal{F}}$, there exists a unique exchangeable distribution $P$ on ordered objects $\mathcal{F}$ for which*

$$Q(\tilde{f}) = \sum_{g \in \tilde{f}} P(g). \tag{11}$$

*Proof.* For existence, set $P(f) := Q(\tilde{f})/|\tilde{f}|$ for $f \in \mathcal{F}$, and note that $g \in \tilde{f} \Rightarrow \tilde{g} = \tilde{f}$. For uniqueness, note that definition 3.1 implies that the restriction of $P$ to any particular class must be uniform, which completely determines $P$. $\square$

We will model real-world permutable objects using an exchangeable model, which will play the role of $P$ in eq. (11). To further simplify our rate expression we will also need the following application of the orbit-stabilizer theorem (see Appendix A for more detail), which is visualized in Figure 1:

**Lemma 3.2.** *Given a permutable class $\mathcal{F}$, for each object $f \in \mathcal{F}$, there is a fixed bijection between the left cosets of $\mathrm{Aut}(\overline{f})$ in $\mathcal{S}_n$ and the isomorphism class $\tilde{f}$. This is induced by the function $\theta_f : \mathcal{S}_n \to \mathcal{F}$ defined by $\theta_f(s) := s \cdot \overline{f}$. This implies that*

$$|\tilde{f}| = \frac{|\mathcal{S}_n|}{|\mathrm{Aut}(f)|} = \frac{n!}{|\mathrm{Aut}(f)|}. \tag{12}$$

*Proof.* Follows directly from the orbit-stabilizer theorem (theorem A.1) and the definitions of $\mathrm{Aut}$, $\overline{f}$ and $\tilde{f}$. $\square$

For any $f \in \mathcal{F}$, this allows us to express the right hand side of eq. (11) as:

$$\sum_{g \in \tilde{f}} P(g) = |\tilde{f}| P(f) = \frac{n!}{|\mathrm{Aut}(f)|} P(f) \tag{13}$$

where the first equality follows from exchangeability of $P$, and the second from eq. (12). Finally, from eqs. (11) and (13), we can immediately write down the following optimal rate expression, which a codec on unordered objects should achieve:

$$\log \frac{1}{Q(\tilde{f})} = \underbrace{\log \frac{1}{P(f)}}_{\text{Ordered rate}} - \underbrace{\log \frac{n!}{|\text{Aut}(f)|}}_{\text{Discount}}. \tag{14}$$

Note that only the $\log 1/P(f)$ term depends on the choice of model. The $\log(n!/|\text{Aut}(f)|)$ term can be computed directly from the data, and is the 'discount' that we get for compressing an *unordered* object vs. compressing an ordered one. The discount is larger for objects which have a smaller automorphism group, i.e. objects which *lack symmetry*. It can be shown that almost all simple graphs have a trivial automorphism group for large enough $n$, see e.g. Bollobás (2001, Chapter 9), and thus in practice the discount is usually equal to or close to $\log n!$.

## 3.1 ACHIEVING THE TARGET RATE FOR UNORDERED OBJECTS

How can we achieve the optimal rate in eq. (14)? In appendix B we give examples of codecs for ordered strings and simple graphs which achieve the 'ordered rate'. To operationalize the negative 'discount' term, we can use the 'bits-back with ANS' method introduced by Townsend et al. (2019), the key idea being to *decode* an ordering as part of an *encode* function (see line 3 in the code below).

The value of the negative term in the rate provides a hint at how exactly to decode an ordering: the discount is equal to the logarithm of the number of cosets of $\text{Aut}(\overline{f})$ in $\mathcal{S}_n$, so a uniform codec for those cosets will consume exactly that many bits. Lemma 3.2 tells us that there is a direct correspondence between the cosets of $\text{Aut}(\overline{f})$ and the set $\tilde{f}$, so if we uniformly decode a choice of coset, we can reversibly map that to an ordering of $f$.

The following is an implementation of shuffle coding, showing, on the right, the effect of the steps on message length.

```
1  def encode(m, f):
2    f_canon = action_apply(canon_perm(f), f)
3    m, s = UniformLCoset(f_canon.aut).decode(m)
4    g = action_apply(s, f_canon)
5    m = P.encode(m, g)
6    return m
7
8  def decode(m):
9    m, g = P.decode(m)
10   s_ = inv_canon_perm(g)
11   f_canon = action_unapply(s_, f)
12   m = UniformLCoset(f_canon.aut).encode(m, s_)
13   return m, f_canon
```

Effect on message length:

$-\log \frac{n!}{|\text{Aut}(f)|}$

$+\log \frac{1}{P(f)}$

The encode function accepts a pair (m, f), and reversibly *decodes* a random choice g from the isomorphism class of f. This is done using a uniform codec for left cosets, UniformLCoset, which we discuss in detail in Appendix C. The canonization on line 2 is necessary so that the decoder can recover the chosen coset and encode it on line 12. While the codec technically maps between $M \times \widetilde{\mathcal{F}}$ and $M$, we avoid representing equivalence classes explicitly as sets, and instead use a single element of the class as a representative. Thus the encoder accepts any f in the isomorphism class being encoded, and the decoder then returns the canonization of f. Similarly, UniformLCoset.encode accepts any element of the coset, and UniformLCoset.decode returns a canonical coset element.

## 3.2 INITIAL BITS

The increase in message length from shuffle coding is equal to the optimal rate in eq. (14). However, the decode step on line 3 of the encode function assumes that there is already some information in the message which can be decoded. At the very beginning of encoding, these 'initial bits' can be

generated at random, but they are unavoidably encoded into the message, meaning that for the first object, the discount is not realized. This constant initialization overhead means that the rate, when compressing only one or a few objects, is not optimal, but tends to the optimal rate if more objects are compressed, as the overhead is amortized.

## 4    RELATED WORK

To date, there has been a significant amount of work on compression of what we refer to as 'ordered' graphs, see Besta and Hoefler (2019) for a comprehensive survey. Compression of 'unordered' graphs, and unordered objects in general, has been less well studied, despite the significant potential benefits of removing order information (see Table 2). The work of Varshney and Goyal (2006) is the earliest we are aware of to discuss the theoretical bounds for compression of sets and multisets, which are unordered strings.

Choi and Szpankowski (2012) discuss the optimal rate for unordered graphs (a special case of our eq. 14), and present a compression method called 'structural ZIP' (SZIP), which asymptotically achieves the rate

$$\log \frac{1}{P_{\mathrm{ER}}(g)} - n \log n + O(n), \tag{15}$$

where $P_{\mathrm{ER}}$ is the Erdős-Rényi $G(n, p)$ model. Compared to our method, SZIP is less flexible in the sense that it only applies to simple graphs (without vertex or edge attributes), and it is not an entropy coding method, thus the model $P_{\mathrm{ER}}$ cannot be changed easily. On the other hand, SZIP can achieve good rates on single graphs, whereas, because of the initial bits issue (see Section 3.2), our method only achieves the optimal rate on *sequences* of objects. We discuss this issue further and provide a quantitative comparison in Section 5.

Steinruecken (2014, 2015, 2016) provides a range of specialized methods for compression of various ordered and unordered permutable objects, including multisets, permutations, combinations and compositions. Steinruecken's approach is similar to ours in that explicit probabilistic modeling is used, although different methods are devised for each kind of object rather than attempting a unifying treatment as we have done.

Our method can be viewed as a generalization of the framework for multiset compression presented in Severo et al. (2023a), which also used 'bits-back with ANS' (BB-ANS; Townsend, 2021; Townsend et al., 2019). Severo et al. (2023a) use interleaving to reduce the initial bits overhead and achieve an optimal rate when compressing a *single* multiset (which can also be applied to a sequence of multisets), whereas the method presented in this paper is optimal only for sequences of unordered objects (including sequences of multisets). However, as mentioned in Section 1, their method only works for multisets and not for more general unordered objects.

There are a number of recent works on deep generative modeling of graphs (see Zhu et al. (2022) for a survey), which could be applied to entropy coding to improve compression rates. Particularly relevant is Chen et al. (2021), who optimize an evidence lower-bound (ELBO), equivalent to an upper-bound on the rate in eq. (14), when $P$ is not exchangeable. Finally, the 'Partition and Code' (PnC; Bouritsas et al., 2021) method uses neural networks to compress unordered graphs. We compare to PnC empirically in Table 3. PnC is also specialized to graphs, although it does employ probabilistic modeling to some extent.

## 5    EXPERIMENTS

To demonstrate the method experimentally, we first applied it to the TUDatasets graphs (Morris et al., 2020), with a very simple Erdős-Rényi $G(n, p)$ model for $P$. Table 2 shows a summary, highlighting the significance of the discount achieved by shuffle coding. We compressed a dataset at a time (note that for each high-level graph type there are multiple datasets in TUDatasets).

To handle graphs with discrete vertex and edge attributes, we treated all attributes as independent and identically distributed (i.i.d.) within each dataset. For each dataset, the codec computes and encodes a separate empirical probability vector for vertices and edges, as well as an empirical $p$ parameter, and the size $n$ of each graph. We use run-length encoding for these meta-data, described

in detail in Appendix E. Some datasets in TUDatasets contain graphs with continuous attributes. We did not encode these attributes, since for these values lossy compression would usually be more appropriate, and the focus of this work is on lossless.

Table 2: For the TUDatasets, this table shows the significance of the discount term - in eq. (14). With an Erdős-Rényi (ER) model, with edge probability adapted to each dataset, the percentage improvement (Discount) is the difference between treating the graph as ordered (Ordered ER) and using Shuffle coding to forget the order (Shuffle coding ER). Rates are measured in bits per edge.

| Graph type | Ordered ER | Shuffle coding ER | Discount |
|---|---|---|---|
| Small molecules | 2.11 | 1.14 | 46% |
| Bioinformatics | 9.20 | 6.50 | 29% |
| Computer vision | 6.63 | 4.49 | 32% |
| Social networks[6] | 3.98 | 2.97 | 26% |
| Synthetic | 5.66 | 2.99 | 47% |

We also compared directly to Bouritsas et al. (2021), who used a more sophisticated neural method to compress graphs (upper part of Table 3). They reported results for six of the datasets from the TUDatasets with vertex and edge attributes removed, and for two of the six they reported results which included vertex and edge attributes. Because PnC requires training, it was evaluated on a random test subset of each dataset, whereas shuffle coding was evaluated on entire datasets.

We found that for some types of graphs, such as the bioinformatics and social network graphs, performance was significantly improved by using a Pólya urn (PU) preferential attachment model for ordered graphs introduced by Severo et al. (2023b). In this model, a sequence of edges is sampled, where the probability of an edge being connected to a specific node is approximately proportional to the number of edges already connected to that node. Such a 'rich-get-richer' dynamic is plausibly present in the formation of many real-world graphs, explaining the urn model's good performance. It treats edges as a set, and we were able to use an inner shuffle codec for sets to encode the edges, demonstrating the straightforward compositionality of shuffle coding. See Appendix D for details. The average initial bit cost per TU dataset in Table 3 is 0.01 bits per edge for both ER and PU, demonstrating good amortization.

As mentioned in Section 4, SZIP achieves a good rate for single graphs, whereas shuffle coding is only optimal for sequences of graphs. In the lower part of Table 3, we compare the 'net rate', which is the increase in message length from shuffle coding the graphs, assuming some existing data is already encoded into the message. The fact that shuffle coding 'just works' with any statistical model for ordered graphs is a major advantage of the method, as demonstrated by the fact that we were easily able to improve on the Erdős-Rényi results by swapping in a recently proposed model.

We report speeds in Appendix F. Our implementation has not yet been optimized. One thing that will not be easy to speed up is canonical ordering, since for this we use the `nauty` and `Traces` libraries, which have already been heavily optimized. Fortunately, those calls are currently only 10 percent of the overall time, and we believe there is significant scope for optimization of the rest.

## 6  LIMITATIONS AND FUTURE WORK

**Time complexity.** Shuffle coding relies on computing an object's canonical ordering and automorphism group, for which no polynomial-time algorithm is known for graphs. In consequence, while `nauty` and `Traces` solve this problem efficiently for various graph classes, it is impractical in the worst case. This limitation can be overcome by approximating an object's canonical ordering, instead of calculating it exactly. This introduces a trade-off between speed and compression rate in the method, and lowers runtime complexity to polynomial time. We leave a detailed description of that more sophisticated method to future work.

**Initial bits.** A limitation of the version of shuffle coding presented in this paper is that it only achieves an optimal rate for sequences; the rate 'discount' cannot be realized in the one-shot case,

---

[6]Three of the 24 social network datasets, REDDIT-BINARY, REDDIT-MULTI-5K, REDDIT-MULTI-12K, were excluded because compression running time was too long.

Table 3: Comparison between shuffle coding, with Erdős-Rényi (ER) and our Pólya urn (PU) models, and the best results obtained by PnC (Bouritsas et al., 2021) and SZIP (Choi and Szpankowski, 2012) for each dataset. We also show the discount realized by shuffle coding. Each SZIP comparison is on a single graph, and thus for shuffle coding we report the optimal (*net*) compression rate, that is the additional cost of compressing that graph assuming there is already some compressed data to append to. All measurements are in bits per edge.

| | | | Shuffle coding | | |
| | Dataset | Discount | ER | PU | PnC |
|---|---|---|---|---|---|
| **Small molecules** | MUTAG | 2.77 | **1.88** | 2.66 | 2.45±0.02 |
| | MUTAG (with attributes) | 2.70 | **4.20** | 4.97 | 4.45 |
| | PTC_MR | 2.90 | **2.00** | 2.53 | 2.97±0.14 |
| | PTC_MR (with attributes) | 2.87 | **4.88** | 5.40 | 6.49±0.54 |
| | ZINC_full | 3.11 | **1.82** | 2.63 | 1.99 |
| **Bioinformatics** | PROTEINS | 2.48 | 3.68 | **3.50** | 3.51±0.23 |
| **Social networks** | IMDB-BINARY | 0.97 | 2.06 | 1.50 | **0.54** |
| | IMDB-MULTI | 0.88 | 1.52 | 1.14 | **0.38** |

| | | | Shuffle coding (net) | | |
| | Dataset | Discount | ER | PU | SZIP |
|---|---|---|---|---|---|
| **SZIP** | Airports (USAir97) | 1.12 | 5.09 | **2.90** | 3.81 |
| | Protein interaction (YeastS) | 3.55 | 6.84 | **5.70** | 7.05 |
| | Collaboration (geom) | 3.45 | 8.30 | **4.41** | 5.28 |
| | Collaboration (Erdos) | 7.80 | 7.00 | **4.37** | 5.08 |
| | Genetic interaction (homo) | 3.97 | 8.22 | **6.77** | 8.49 |
| | Internet (as) | 7.34 | 8.37 | **4.47** | 5.75 |

as explained in Section 3.2. However, it is possible to overcome this by interleaving encoding and decoding steps, as done in the 'bit-swap' method of Kingma et al. (2019). Information can be eagerly encoded during the progressive decoding of the coset, reducing the initial bits needed by shuffle coding from $O(\log n!)$ to $O(\log n)$. This is a generalization of the multiset coding method described by Severo et al. (2023a). We again defer a detailed description to future work.

**Models.** Unlike PnC, we do not rely on compute-intensive learning or hyperparameter tuning. Shuffle coding achieves state-of-the-art compression rates when using simple models with minimal parameters. There is currently active research on deep generative models for graphs, see Zhu et al. (2022) for a survey. We expect improved rates for shuffle coding when combined with such neural models.

## 7 CONCLUSION

A significant proportion of the data which needs to be communicated and stored is fundamentally unordered. We have presented shuffle coding, the first general method which achieves an optimal rate when compressing sequences of unordered objects. We have also implemented experiments which demonstrate the practical effectiveness of shuffle coding for compressing many kinds of graphs, including molecules and social network data. We look forward to future work applying the method to other forms of unordered data, and applying more sophisticated probabilistic generative models to gain improvements in compression rate.

ACKNOWLEDGMENTS

James Townsend acknowledges funding under the project VI.Veni.212.106, financed by the Dutch Research Council (NWO). We thank Ashish Khisti for discussions and encouragement, and Heiko Zimmermann for feedback on the paper.

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

## A  GROUP ACTIONS, ORBITS AND STABILIZERS

This appendix gives the definitions of group actions, orbits and stabilizers as well as a statement and proof of the orbit-stabilizer theorem, which we make use of in section 3. We use the shorthand $H \leq G$ to mean that $H$ is a subgroup of $G$, and for $g \in G$, we use the usual notation, $gH := \{gh \mid h \in H\}$ and $Hg := \{hg \mid h \in H\}$ for left and right cosets, respectively.

**Definition A.1** (Group action). For a set $X$ and a group $G$, a *group action*, or simply *action*, is a binary operator

$$\cdot_G : G \times X \to X \tag{16}$$

which respects the structure of $G$ in the following sense:

1. The identity element $e \in G$ is neutral, that is $e \cdot_G x = x$.

2. The operator $\cdot_G$ respects composition. That is, for $g, h \in G$,

$$g \cdot_G (h \cdot_G x) = (gh) \cdot_G x. \tag{17}$$

We will often drop the subscript $G$ and use infix $\cdot$ alone where the action is clear from the context.

**Definition A.2** (Orbit). An action of a group $G$ on a set $X$ induces an equivalence relation $\sim_G$ on $X$, defined by

$$x \sim_G y \quad \text{if and only if there exists} \quad g \in G \quad \text{such that} \quad y = g \cdot x. \tag{18}$$

We refer to the equivalence classes induced by $\sim_G$ as *orbits*, and use $\mathrm{Orb}_G(x)$ to denote the orbit containing an element $x \in X$. We use $X/G$ to denote the set of orbits, so for each $x \in X$, $\mathrm{Orb}_G(x) \in X/G$.

**Definition A.3** (Stabilizer subgroup). For an action of a group $G$ on a set $X$, for each $x \in X$, the *stabilizer*

$$\mathrm{Stab}_G(x) := \{g \in G \mid g \cdot x = x\} \tag{19}$$

forms a subgroup of $G$.

We make use of the orbit-stabilizer theorem in Section 3. Here we give a statement and brief proof of this well-known theorem.

**Theorem A.1** (Orbit-stabilizer theorem). *For an action of a finite group $G$ on a set $X$, for each $x \in X$, the function $\theta_x \colon G \to X$ defined by*

$$\theta_x(g) := g \cdot x \tag{20}$$

*induces a bijection from the left cosets of $\mathrm{Stab}_G(x)$ to $\mathrm{Orb}_G(x)$. This implies that the orbit $\mathrm{Orb}_G(x)$ is finite and*

$$|\mathrm{Orb}_G(x)| = \frac{|G|}{|\mathrm{Stab}_G(x)|}. \tag{21}$$

*Proof.* We show that $\theta_f$ induces a well defined function on the left-cosets of $\mathrm{Stab}_G(x)$, which we call $\tilde{\theta}_f$. Specifically, we define

$$\tilde{\theta}_f(g \, \mathrm{Stab}_G(x)) := g \cdot x, \tag{22}$$

and show that $\tilde{\theta}_f$ is injective and surjective.

To see that $\tilde{\theta}_f$ is well defined and injective, note that

$$h \in g \, \mathrm{Stab}_G(x) \iff g^{-1}h \in \mathrm{Stab}_G(x) \tag{23}$$
$$\iff g^{-1}h \cdot x = x \tag{24}$$
$$\iff g \cdot x = h \cdot x, \tag{25}$$

using the definition of $\mathrm{Stab}_G$.

For surjectivity, we have

$$y \in \mathrm{Orb}_G(x) \implies \exists g \in G \text{ s.t. } y = g \cdot x \tag{26}$$
$$\implies y = \tilde{\theta}_f(g \, \mathrm{Stab}_G(x)) \tag{27}$$

using the definition of $\mathrm{Orb}_G$. $\qquad\square$

In Appendix C.2, it will be helpful to have an explicit bijection between $G$ and the Cartesian product $\mathrm{Orb}_G(x) \times \mathrm{Stab}_G(x)$. This requires a way of selecting a canonical element from each left coset of $\mathrm{Stab}_G(x)$ in $G$. This is similar to the canonical ordering of definition 2.4:

**Definition A.4** (Transversal). For a group $G$ with subgroup $H \leq G$, a *transversal* of the left cosets of $H$ in $G$ is a mapping $t : G \to G$ such that

1. For all $g \in G$, we have $t(g) \in gH$.

2. For all $f, g \in G$, if $f \in gH$, then $t(f) = t(g)$.

Given such a transversal, we can setup the bijection mentioned above:

**Lemma A.1.** *Let $G$ be a group acting on a set $X$. If, for $x \in X$, we have a transversal $t_x$ of the left cosets of $\mathrm{Stab}_G(x)$ in $G$, then we can form an explicit bijection between $G$ and $\mathrm{Orb}_G(x) \times \mathrm{Stab}_G(x)$.*

*Proof.* For $g \in G$, let

$$o_x(g) := g \cdot x \tag{28}$$

$$s_x(g) := t_x(g)^{-1}g, \tag{29}$$

then $o_x \in \mathrm{Orb}_G(x)$. By condition 1 in definition A.4, there exists $h \in \mathrm{Stab}_G(x)$ such that $t(g) = gh$, and in particular $s_x(g) = h \in \mathrm{Stab}_G(x)$. So there is a well-defined function $\phi_x(g) := (o_x(g), s_x(g))$ with $\phi_x : G \to \mathrm{Orb}_G(x) \times \mathrm{Stab}_G(x)$.

To see that $\phi_x$ is injective, suppose that $\phi_x(f) = \phi_x(g)$. Then $o_x(f) = o_x(g)$, so $f \cdot x = g \cdot x$, and therefore $f \in g\,\mathrm{Stab}_G(x)$. Condition 2 in definition A.4 implies that $t_x(f) = t_x(g)$, and since $s_x(g) = s_x(f)$ we have $t_x(f)^{-1}f = t_x(g)^{-1}g$, so $f = g$.

The orbit-stabilizer theorem implies that $|G| = |\mathrm{Orb}_G(x)||\mathrm{Stab}_G(x)|$, and therefore if $\phi_x$ is injective it must also be bijective. $\qquad\square$

## B    CODECS FOR ORDERED OBJECTS

Codecs for strings and graphs can be composed from the primitive codecs introduced in section 2.2:

```python
def String(ps, length):
  def encode(m, string):
    assert len(string) == length
    for c in reversed(string):
      m = Categorical(ps).encode(m, c)
    return m

  def decode(m):
    string = []
    for _ in range(length):
      m, c = Categorical(ps).decode(m)
      string.append(c)
    return m, str(string)
  return Codec(encode, decode)
```

```python
def ErdosRenyi(n, p):
  def encode(m, g):
    assert len(g) == n
    for i in reversed(range(n)):
      for j in reversed(range(i)):
        e = g[i][j]
        m = Bernoulli(p).encode(m, e)
    return m

  def decode(m):
    g = []
    for i in range(n):
      inner = []
      for j in range(i)
        m, e = Bernoulli(p).decode(m)
        inner.append(e)
      g.append(inner)
    return (m, g)
  return Codec(encode, decode)
```

Left: Codec for fixed-length strings implemented by applying a `Categorical` codec to each character. Right: Codec for graphs respecting an Erdős-Rényi distribution $G(n, p)$, implemented by applying the `Bernoulli` codec to each edge.

## C    A UNIFORM CODEC FOR COSETS OF A PERMUTATION GROUP

Shuffle coding, as described in Section 3, requires that we can encode and decode left cosets in $\mathcal{S}_n$ of the automorphism group of a permutable object. In this appendix we describe a codec for cosets

of an *arbitrary* permutation group characterized by a list of generators. We first describe the codec, which we call `UniformLCoset`, on a high level and then in Appendices C.1 and C.2, we describe the two main components in more detail.

The optimal rate for a uniform coset codec is equal to the log of the number of cosets, that is

$$\log \frac{|\mathcal{S}_n|}{|H|} = \log n! - \log|H|. \tag{30}$$

This rate expression hints at an encoding method: to encode a coset, we first decode a choice of element of the coset (equivalent to decoding a choice of element of $H$ and then multiplying it by a canonical element of the coset), and then encode that chosen element using a uniform codec on $\mathcal{S}_n$. Note that if the number of cosets is small we could simply encode the index of the coset directly, but in practice this is rarely feasible.

The following is a concrete implementation of a left coset codec:

```
def UniformLCoset(grp):                          Effects on  l(m):
  def encode(m, s):
    s_canon = coset_canon(grp, s)
    m, t = UniformPermGrp(grp).decode(m)         − log|H|
    u = s_canon * t
    m = UniformS(n).encode(m, u)                 + log(n!)
    return m

  def decode(m):
    m, u = UniformS(n).decode(m)
    s_canon = coset_canon(subgrp, u)
    t = inv(s_canon) * u
    m = UniformPermGrp(grp).encode(m, t)
    return m, s_canon
  return Codec(encode, decode)
```

The codecs `UniformS` and `UniformPermGrp` are described in Appendix C.1 and Appendix C.2 respectively. `UniformS(n)` is a uniform codec over the symmetric group $\mathcal{S}_n$, and `UniformPermGrp` is a uniform codec over elements of a given permutation group, i.e., a subgroup of $\mathcal{S}_n$.

We use a *stabilizer chain*, discussed in more detail in Appendix C.2, which is a computationally convenient representation of a permutation group. A stabilizer chain allows computation of a transversal which can be used to canonize coset elements (line 3 and line 11 in the code above). Routines for constructing and working with stabilizer chains are standard in computational group theory, and are implemented in SymPy (`https://www.sympy.org/`), as well as the GAP system (`https://www.gap-system.org/`), see Holt (2005, Chapter 4) for theory and description of the algorithms. The method we use for `coset_canon` is implemented in the function `MinimalElementCosetStabChain` in the GAP system.

## C.1 A UNIFORM CODEC FOR PERMUTATIONS IN THE SYMMETRIC GROUP

We use a method for encoding and decoding permutations based on the Fisher-Yates shuffle (Knuth, 1981, pp. 139–140). The following is a Python implementation:

```
def UniformS(n):
  def swap(s, i, j):
    si_old = s[i]
    s[i] = s[j]
    s[j] = si_old

  def encode(m, s):
    p = list(range(n))
    p_inv = list(range(n))
    to_encode = []
```

```
11        for j in reversed(range(2, n + 1)):
12          i = p_inv[x[j - 1]]
13          swap(p_inv, p[j - 1], x[j - 1])
14          swap(p, i, j - 1)
15          to_encode.append(i)
16
17        for j, i in zip(range(2, n + 1), reversed(to_encode)):
18          m = Uniform(j).encode(m)
19        return m
20
21      def decode(m):
22        s = list(range(n))
23        for j in reversed(range(2, n + 1)):
24          m, i = Uniform(j).decode(m)
25          swap(s, i, j - 1)
26        return m, s
27      return Codec(encode, decode)
```

The decoder closely resembles the usual Fisher-Yates sampling method, and the encoder has been carefully implemented to exactly invert this process. Both encoder and decoder have time complexity in $O(n)$.

### C.2 A UNIFORM CODEC FOR PERMUTATIONS IN AN ARBITRARY PERMUTATION GROUP

For coding permutations from an arbitrary permutation group, we use the following construction, which is a standard tool in computational group theory (see Holt (2005) and Seress (2003)):

**Definition C.1** (Base, stabilizer chain). Let $H \leq \mathcal{S}_n$ be a permutation group, and $B = (b_0, \ldots, b_{K-1})$ a list of elements of $[n]$. Let $H_0 := H$, and $H_k := \mathrm{Stab}_{H_{k-1}}(b_{k-1})$ for $k = 1, \ldots, K$. If $H_K$ is the trivial group containing only the identity, then we say that $B$ is a *base* for $H$, and the sequence of groups $H_0, \ldots, H_K$ is a *stabilizer chain* of $H$ relative to $B$.

Bases and stabilizer chains are guaranteed to exist for all permutation groups, and can be efficiently computed using the Schreier-Sims algorithm (Sims, 1970). The algorithm also produces a transversal for the left cosets of each $H_{k+1}$ in $H_k$ for each $k = 0, \ldots, K-1$, in a form known as a Schreier tree (Holt, 2005).

If we define $O_k := \mathrm{Orb}_{H_k}(b_k)$, for $k = 0, \ldots, K-1$, then by applying the orbit-stabilizer theorem recursively, we have $|H| = \prod_{k=0}^{K-1} |O_k|$, which gives us a decomposition of the optimal rate that a uniform codec on $H$ should achieve:

$$\log|H| = \sum_{k=0}^{K-1} \log|O_k|. \tag{31}$$

Furthermore, by applying lemma A.1 recursively, using the transversals produced by Schreier-Sims, we can construct an explicit bijection between $H$ and the Cartesian product $\prod_{k=0}^{K-1} O_k$. We use this bijection, along with a sequence of uniform codecs on $O_0, \ldots, O_{K-1}$ for coding automorphisms at the optimal rate in eq. (31). For further details refer to the implementation.

## D PÓLYA URN MODEL DETAILS

We implemented Pólya urn models mostly as described in Severo et al. (2023b), with few modifications. Differently to the original implementation, we apply shuffle coding to the list of edges, resulting in a codec for the set of edges.

We also disallow edge redraws and self-loops, leading to an improved rate, as shown in appendix G. This change breaks edge-exchangeability, leading to a 'stochastic' codec, meaning that the code length depends on the initial message. Shuffle coding is compatible with such models. In this more general setting, the ordered log-likelihood term in the optimal rate (eq. 14) is replaced with a variational 'evidence lower bound' (ELBO). The discount term is unaffected. The derivations in the

main text are based on the special case of exchangeable models, where log-likelihoods are exact, for simplicity. They can be generalized with little effort and new insight.

## E  PARAMETER CODING DETAILS

All bit rates reported for our experiments include model parameters. Once per dataset, we code the following lists of natural numbers by coding both the list length and the bit count $\lceil \log m \rceil$ of the maximum element $m$ with a 46-bit and 5-bit uniform codec respectively, as well as each element of the list with a codec respecting a log-uniform distribution in $[0, \lceil \log m \rceil]$:

- A list resulting from sorting the graphs' numbers of vertices, and applying run-length coding, encoding run lengths and differences between consecutive numbers of vertices.
- For datasets with vertex attributes: a list of all vertex attribute counts within a dataset.
- For datasets with edge attributes: a list of all edge attribute counts within a dataset.
- For Erdős-Rényi models: a list consisting of the following two numbers: the total number of edges in all graphs, and the number of vertex pairs that do not share an edge.

Coding these empirical count parameters allows coding the data according to maximum likelihood categorical distributions. For Pólya urn models, we additionally code the edge count for each graph using a uniform codec over $[0, \frac{1}{2}n(n-1)]$, exploiting the fact that the vertex count $n$ is already coded as described above. For each dataset, we use a single bit to code whether or not self-loops are present and adapt the codec accordingly.

## F  COMPRESSION SPEED

We show compression and decompression speeds of our experiments in Table 4. These speeds include time needed for gathering dataset statistics and parameter coding. The results show that for our implementation, only a small fraction of runtime is spent on finding automorphism groups and canonical orderings with `nauty`.

## G  MODEL ABLATIONS

We present results of additional ablation experiments on the PnC datasets in Table 5. We do an ablation that uses a uniform distribution for vertex and edge attributes with an Erdős-Rényi model (unif. ER). There is a clear advantage to coding maximum-likelihood categorical parameters (ER), justifying it as the approach used throughout this paper. We also show the rates obtained by the original method proposed in Severo et al. (2023b) (PU redr.), demonstrating a clear rate advantage of our approach disallowing edge redraws and self-loops (PU) in the model.

Table 4: Compression and decompression speeds in kilobytes per second (kB/s) of shuffle coding with the Erdős-Rényi (ER) and Pólya urn (PU) models, for all previously reported TU and SZIP datasets. We show SZIP compression speeds calculated from the runtimes reported in Choi and Szpankowski (2012) for comparison. All results are based on the ordered ER rate as the reference uncompressed size. All shuffle coding speeds are for a single thread on a MacBook Pro 2018 with a 2.7GHz Intel Core i7 CPU. We also report the share of time spent on `nauty` calls that determine the canonical ordering and generators of the automorphism group of all graphs.

| | | ER | | PU | | SZIP |
|---|---|---|---|---|---|---|
| Dataset | nauty | encode | decode | encode | decode | encode |
| **TU by type** | | | | | | |
| Small molecules | 15% | 54 | 56 | – | – | – |
| Bioinformatics | 2% | 51 | 66 | – | – | – |
| Computer vision | 3% | 25 | 28 | – | – | – |
| Social networks | <1% | 0.440 | 0.467 | – | – | – |
| Synthetic | 7% | 98 | 110 | – | – | – |
| **Small molecules** | | | | | | |
| MUTAG | 7% | 115 | 122 | 51 | 40 | – |
| MUTAG (with attributes) | 16% | 135 | 141 | 67 | 62 | – |
| PTC_MR | 8% | 107 | 103 | 50 | 45 | – |
| PTC_MR (with attributes) | 18% | 117 | 125 | 67 | 62 | – |
| ZINC_full | 7% | 105 | 105 | 50 | 47 | – |
| **Bioinformatics** | | | | | | |
| PROTEINS | 3% | 88 | 94 | 30 | 30 | – |
| **Social networks** | | | | | | |
| IMDB-BINARY | 4% | 17 | 18 | 8 | 8 | – |
| IMDB-MULTI | 3% | 11 | 12 | 6 | 5 | – |
| **SZIP** | | | | | | |
| Airports (USAir97) | 1% | 82 | 78 | 5 | 5 | 164 |
| Protein interaction (YeastS) | 8% | 2.442 | 2.391 | 1.238 | 0.859 | 77 |
| Collaboration (geom) | <1% | 0.004 | 0.005 | 0.005 | 0.005 | 64 |
| Collaboration (Erdos) | 15% | 0.025 | 0.025 | 0.024 | 0.024 | 18 |
| Genetic interaction (homo) | 7% | 0.180 | 0.154 | 0.117 | 0.141 | 32 |
| Internet (as) | 13% | 0.002 | 0.003 | 0.002 | 0.002 | 7 |

Table 5: Model ablations compared to PnC. All results are in bits per edge.

| | Shuffle coding | | | | |
|---|---|---|---|---|---|
| Dataset | unif. ER | ER | PU | PU redr. | PnC |
| **Small molecules** | | | | | |
| MUTAG | – | **1.88** | 2.66 | 2.81 | 2.45±0.02 |
| MUTAG (with attributes) | 6.37 | **4.20** | 4.97 | 5.13 | 4.45 |
| PTC_MR | – | **2.00** | 2.53 | 2.74 | 2.97±0.14 |
| PTC_MR (with attributes) | 8.04 | **4.88** | 5.40 | 5.61 | 6.49±0.54 |
| ZINC_full | – | **1.82** | 2.63 | 2.75 | 1.99 |
| **Bioinformatics** | | | | | |
| PROTEINS | – | 3.68 | **3.50** | 3.62 | 3.51±0.23 |
| **Social networks** | | | | | |
| IMDB-BINARY | – | 2.06 | 1.50 | 2.36 | **0.54** |
| IMDB-MULTI | – | 1.52 | 1.14 | 2.17 | **0.38** |

