# OpenReview forum: "Entropy Coding of Unordered Data Structures"
_ICLR.cc/2024/Conference — ICLR 2024 poster_

### Official Review · Reviewer_72sp · 2023-10-19

**Soundness:** 3 good
**Presentation:** 2 fair
**Contribution:** 3 good
**Rating:** 8
**Confidence:** 4

**Summary:**

This paper proposes a general method called shuffle coding for compressing sequences of unordered objects using bits-back coding.
The proposed shuffle coding is applicable to multisets, graphs, hypergraphs etc.
SOTA compression rate is achieved on several graph datasets including molecular data.

**Strengths:**

The problem of compressing unordered data optimally by considering the storing order as redundancy is novel to me.
The paper is a solid contribution with solid theoretical foundations. The mathematical structures from group theory are closely connected with the target unordered data.
The idea of decoding the ordering as part of an encode function in the spirits of bits-back coding is very interesting.
The background and proposed methods are clearly explained and relatively easy to follow.
Experiments are conducted on server different datasets, and achieve sota compression rate.
Limitations of the proposed method is properly discussed.

**Weaknesses:**

One major concern is the practical value of compressing those unordered data. Compared with images or videos, it seems to me the data volume of those unordered date is very limited. However, I still appreciate the technical contribution of this paper.

Currently the numbers in Table 4 is relatively not easy to understand. It would be better to report speed as GB/s or MB/s and compare this with previous methods, so that readers can better feel the efficiency.

**Questions:**

Definition 2.5 is described in a very concise way. It is not clear why the definition is specifically connected with rANS. I think is also applies to arithmetic coding.

It is not clear how eq9 is related to the real bitrate. In this definition, the data is compressed in an instance-wise manner and l(m) is the initial bit cost which can be amortized later if we compress more instances. However, log(1/p(x)), which is defined as the rate of the codec, is an averaged value averaging over the input symbol number. I do not understand why l(m) can be added with log(1\p(x)).

Definition 2.3 (Automorphism group) is actually the difinition of stabilizer. If I remember correctly, the concept of Automorphism in group theory is connected with operation by conjugation. Did I miss anything?

---

> ### Author Response · Authors · 2023-11-15
>
> We thank reviewer 72sp for their review.
>
> > Currently the numbers in Table 4 is relatively not easy to understand. It would be better to report speed as GB/s or MB/s and compare this with previous methods, so that readers can better feel the efficiency.
>
> We fully agree with the reviewer, and updated Table 4 to show speeds in the more standard unit of kB/s. While we did not find information on speeds for PnC, we added a column for SZIP compression speeds for comparison (no decompression speeds were reported).
>
> ---
>
> > Definition 2.5 is described in a very concise way. It is not clear why the definition is specifically connected with rANS. I think is also applies to arithmetic coding.
>
> Our method requires stack-like (LIFO) codecs, such as those based on the range variant of asymmetric numeral systems (rANS), to recover bits corresponding to the redundant order using bits-back. Queue-like (FIFO) codes such as arithmetic coding cannot be directly used to implement bits-back coding, see Townsend et al. (2019) for more detail. This was previously not explicitly stated. We revised section 2.2, including definition 2.5, to clarify this.
>
> ---
>
> > It is not clear how eq9 is related to the real bitrate. In this definition, the data is compressed in an instance-wise manner and l(m) is the initial bit cost which can be amortized later if we compress more instances. However, log(1/p(x)), which is defined as the rate of the codec, is an averaged value averaging over the input symbol number. I do not understand why l(m) can be added with log(1\p(x)).
>
> If $m$ is the initial message, the left-hand side $l(\text{encode}(m, x))$ of eq. 9 is the real bit rate for a specific input $x$, measuring how many bits are needed to store $x$ without encoding any other information. $\log(1/\text{p}(x))$ is the ideal (amortized) bit rate for that specific input $x$, bounding the real bit rate from below. This is not an average over all possible inputs $x$. Note that in general, $m$ can contain previously compressed objects, and $l(m)$ is therefore not necessarily just the initial bit cost.
>
> ---
>
> > Definition 2.3 (Automorphism group) is actually the difinition of stabilizer. If I remember correctly, the concept of Automorphism in group theory is connected with operation by conjugation. Did I miss anything?
>
> That is all absolutely correct. The automorphism group of $f$ is the stabilizer subgroup of $f$ under the action of $\mathcal{S}_n$, as mentioned right after definition 2.3. The concepts of automorphism and conjugation are indeed closely related in group theory. For example, the function $f_h(g) = h^{-1}gh$ that conjugates any element $g$ of a group $G$ by a fixed element $h$ of $G$ is an automorphism of $G$ (called an inner automorphism).

---

> > ### Comment · Reviewer_72sp · 2023-11-18
> > **Post rebuttal**
> >
> > Thanks for the reply. I keep my score

---

### Official Review · Reviewer_NBHy · 2023-10-20

**Soundness:** 3 good
**Presentation:** 3 good
**Contribution:** 3 good
**Rating:** 6
**Confidence:** 3

**Summary:**

## Summary
* This paper propose shuffle coding, a BB-ANS approach towards lossless compression of unordered sets. The scenario described by the paper seems to related to the Birkhoff Polytope in variational inference of permutation [Linderman 2017, Reparameterizing the Birkhoff Polytope for Variational Permutation Inference], where each permutation's likelihood is the same. The authors demonstrate the effectiveness of their approach on various datasets.

**Strengths:**

## Strength
* The unordered set / graph compression problem is of good practical value. The proposed approach is a neat extension of bits-back coding. It is simple, novel and works well.

**Weaknesses:**

## Weakness
* As the authors have discussed, the current initial bits required is quite large. This hinders the practical application of the proposed approach to one-shot object coding. Though it is still possible to apply this approach to a dataset to amortize the initial bits. An alternative to the bit-swap approach mentioned by authors is correlation communication [Harsha 2010, The Communication Complexity of Correlation] [Li 2018, Strong Functional Representation Lemma and Applications to Coding Theorems] [Theis 2021, Algorithms for the Communication of Samples], which has extra overhead of approximately $\log \log N!$, but does not require any initial bits.

**Questions:**

## Questions
* For practically large structure, what is the computational cost for finding and sampling from the isomorphism class $\hat{f}$? For practical codec, the metrics such as latency and throughput should also be discussed.

---

> ### Author Response · Authors · 2023-11-15
>
> We thank reviewer NBHy for taking the time to review our paper.
>
> > The scenario described by the paper seems to related to the Birkhoff Polytope in variational inference of permutation
>
> Indeed, as the reviewer pointed out, our method is related to [Linderman 2017], where they reparameterize the Birkhoff Polytope to perform variational inference over permutation matrices. The vertices of the n-dimensional Birkhoff Polytope represent permutations over n elements. Our decoding algorithm can be seen as performing inference of the permutation applied to the input via canonicalization. Relaxing this inference to the inner hull of the Birkhoff Polytope, as done in [Linderman 2017], could be a way to perform lossy compression of combinatorial objects.
>
> ---
>
> > An alternative to the bit-swap approach mentioned by authors is correlation communication
>
> While correlation communication methods could theoretically be an alternative to the future work of interleaving, these methods have computational complexity which scale exponentially with the entropy of the source, making them impractical. However, the reviewer is correct that these methods could in theory be used to avoid the initial bits problem, if we disregard computational complexity.
>
> These methods relate to bits-back coding in the following way. Consider the source $X$ and another random variable $Y$ from which we can recover $X$ deterministically, i.e., $X = f(Y)$ for some function $f$. Note this implies that $H(X|Y) = 0$. As a concrete example from BB-ANS, $Y = (Z, X)$ where $Z$ is the latent variable of the VAE. Then, bits-back coding implements rate $$R = H(Y) - H(Y|X) = I(X; Y) = H(X) - H(X|Y) = H(X),$$ where $-H(Y|X)$ is the savings from the bits-back step.
>
> Correlation communication methods can be used to communicate a sample Y | X, achieving a rate of $R + log(R+1) + 5$ via the Poisson Functional Representation Lemma (in this case, the discrete version which falls back onto the exponential races) [Li & El Gammal 2017], and X can then be recovered from that sample. Unfortunately, as mentioned, the computational complexity would be much larger than our method.
>
> Bits-back can be viewed as an extremely efficient algorithm for “deterministic” correlation communication (i.e., when $X = f(Y)$ for some $f$).
>
> ---
>
> > For practically large structure, what is the computational cost for finding and sampling from the isomorphism class $\tilde{f}$?
>
> We provided relative timings for nauty, the library we used to find the canonical labeling and automorphism group (these are the computational bottlenecks for sampling from the isomorphism class), in Appendix E. As mentioned in the paper, no polynomial-time algorithm for the graph isomorphism is known, but nauty and Traces solve this problem efficiently for various graph classes. Thorough benchmarks of nauty and traces on different graph types and sizes can be found at https://pallini.di.uniroma1.it/.
>
> ---
>
> > For practical codec, the metrics such as latency and throughput should also be discussed.
>
> We have updated Appendix E to now report compression throughput. We believe that latency is more appropriate for streaming applications, such as video decoding. We would envision our codec being used in a non-streaming setting, where a whole file would be downloaded and then decompressed.

---

> > ### Comment · Reviewer_NBHy · 2023-11-22
> >
> > Thanks for the feedback, I keep my rating as accept.

---

### Official Review · Reviewer_aWcE · 2023-10-30

**Soundness:** 3 good
**Presentation:** 3 good
**Contribution:** 3 good
**Rating:** 6
**Confidence:** 3

**Summary:**

In this work, the authors introduce _shuffle coding_, a method for compressing sequences of unordered objects based on bits-back coding. They provide an exposition of the group-theoretic fundamentals relevant for shuffle coding (Section 2 and Appendix A), define the desiderata required for an optimal-rate codec for unordered sequences (Section 3) and present shuffle coding, an algorithm that fulfils these desiderata (Section 3.1). They apply shuffle coding to unordered data structures (specifically graphs), showing strong performance compared to PnC and SZIP (Section 5).

**Strengths:**

This paper presents a few key strengths which, in my view, are as follows:

__Elegant unified framework:__
This paper provides an unified theoretical framework for compressing unordered objects, such as multisets and graphs.
This approach is based on the elegant idea that the order of the parts of an object does not matter, one can reduce the cost of communicating the object by getting a certain number of bits, i.e. the bits corresponding to a particular ordering of the parts, back.
This generality is appealing because one does not have to devise specialised method for each different type of unstructured object.
However, it should be noted that this framework assumes access to a "canonicalisation" function, which determines a canonical representation  as well as the automorphism group of the object (e.g. graph) in question (also see weaknesses section below).

__Strong compression performance:__
The authors demonstrate that shuffle coding achieves strong compression rates in practice.
In particular, shuffle coding outperforms SZIP and PnC on a range of graph datasets.
This is an encouraging result because, while the proposed method is asymptotically optimal, it also involves a non-trivial overhead (which is amortised as more data are compressed).
Therefore it is good that despite this cost, shuffle coding seems to perform well in practice.

__Well written paper:__
The paper is generally clearly written and well motivated.
The authors have made an effort to discuss the limitations of their work, specifically about the time complexity and initial rate overheads induced by their method.
I appreciated the illustrations and pseudocode listings in the main text, which helped understand their method.

**Weaknesses:**

The paper's main weaknesses, in my view, revolve around the practical applicability of shuffle coding:

__Large runtime complexity:__
As the authors note, applying shuffle coding to a graph requires solving a graph isomorphism problem, for which no polynomial-time algorithm is known.
This can be a significant hurdle when coding larger graphs.
The authors brought up this issue in the paper, and suggested that approximately solving the isomorphism problem is a promising way to scale the method.
However, in its current form, the method does not scale to larger graphs.
Further, it is unclear what the tradeoff between the communication rate and computational complexity of the method would be, if one were to use an approximate scheme instead.

__Initial bit overhead:__
Due to the requirement of initial bits, shuffle coding introduces a communication overhead to the transmitted message.
This can be significant in the one-shot or few-shot case, making the method from compressing small sequences of messages.
I think the authors should specify the amount of extra bits used in the experiments (see Table 3) in the main text.

__Contribution as a general solution seems somewhat exaggerated:__
While the shuffle coding approach is general from a theoretical point of view, it requires access to a "canonicalisation" function.
The authors focus on graphs, for which existing libraries offering this functionality exist.
For other permutable classes, the authors argue that one can "embed objects into graphs in such a way that the structure is preserved and the canonization remains valid."
However, it is unclear whether, for example, this embedding might affect the compression rate, or how difficult it is to construct such an embedding.
In light of this latter point, the statement that the authors' "implementation can easily be adapted to different data types" in the abstract may be regarded as exaggerated (if for example coming up with an algorithm that constructs such graph embeddings is challenging, or if the computational / memory complexity of such an algorithm is large).
A more measured statement in the appendix and / or positioning of the main text might be beneficial in this regard.

__Summary:__
In conclusion, while I find the paper to be well-motivated and elegant, I think the practical applicability of shuffle coding may be limited due to runtime issues, as well as issues pertaining to the embedding of other kinds of permuted classes on graphs, and the initial bit overhead.
Therefore I have recommended a moderately positive score for the paper, but I am willing to adjust my score if the authors address my points of concern raised above, and the questions and recommendations made below.

**Questions:**

Below are some questions for the authors and suggestions that I think could improve the paper.

## Questions

__Clarification on the discount factors for Table 3:__
The ER and PU models are specific probabilistic models for graphs and, as the authors explain, they can be swapped in with any other exchangeable model of graphs.
Therefore, while it is important to report the overall communication cost in the experiments, an equally important (in my view) quantity to report is the rate discount afforded by shuffle coding.
What are the discounts corresponding to the results in table 3?
Perhaps the authors can report these as an extra column, since the discount is a function of the graph alone and not the modelling distribution $P.$


__Extending the framework with approximate isomorphism solutions:__
The authors claim that while no known polynomial-time solution for solving the exact graph isomorphism problem is known

> this limitation can be overcome by approximating an object’s canonical ordering, instead of calculating it exactly. This introduces a trade-off between speed and compression rate in the method, and lowers runtime complexity to polynomial time.

Can the authors comment on the tradeoff between the speed and compression rate of this modified method, here and / or the main text?


## Suggestions

__Definition 2.5:__
I think that definition 2.5 is not totally accurate and / or could be improved.
Presumably, what the authors mean by "inverse functions" is that $\texttt{decode}$ is the inverse function of $\texttt{encode}.$
In that case the $\texttt{decode}$ function isn't really needed and / or can be defined as $\texttt{decode} = \texttt{encode}^{-1}.$
Also, the statement "with respect to $P$" does not seem to make sense on its own.
I think clearer statement is to say that

> an optimal codec for $P$ is an invertible function $\texttt{encode} : M \times X \to M$, which is optimal with respect to $P,$ in the sense that for any [...].

More importantly however, in shuffle coding, the $\texttt{encode}$ function is _not invertible_, because for $f \in \mathcal{F},$ applying $\texttt{decode}(\texttt{encode}(M, f))$ returns the message $M$ together with the canonical representation $\bar{f}$ rather than $f$ itself (see for example the function signature and return statements in the code listing in Section 3.1).
While this is a technicality that does not seem affect the validity of the algorithm, I think the authors should modify their definition and / or exposition to account for this.

__Framing of definition 2.5:__
I think definition 2.5 and the paragraph under it could be framed better.
In particular, the authors define optimal codecs and explain that "definition 2.5 captures the abstract properties of codecs based on the range variant of asymmetric numeral systems."
I think a clearer approach would be to say up-front that their plan is to set up a method that gets those bits which correspond to permutations of the graph in question back.
This in turn necessitates using a last-in-first-out codec, such as (r)ANS, and that definition 2.5 specifies "Optimal last-in-first-out codecs", rather than "Optimal codecs".

__Initial bit overhead statement:__
The authors argue that the constant bit overhead incurred by bits-back "exists for all entropy coding methods."
While all entropy coding methods have a constant overhead that is amortised as more and more data are compressed, typical methods such as Arithmetic Coding (AC), have constant overheads of the order of a couple of bits.
This is far smaller than the claimed overhead of "at most 64 bits" of bits-back for shuffle coding.
I think the authors' phrasing somewhat misrepresents the overhead of other entropy coding methods.
A more accurate statement would be welcome here, such as: "As in other entropy coding methods, which invariably have similar (though typically smaller) constant overheads, the constant 'initial bit cost' of bits-back is amortised as more data are compressed."

__All units in bits?__
Are all reported rates, including for example Table 1, in bits?
If so, it would be good to add a statement up front, early in the paper, that specifies this.

---

> ### Author Response · Authors · 2023-11-15
>
> We thank reviewer aWcE for their thoughtful, thorough and constructive review.
>
> ## Weaknesses
> **Large runtime complexity:** We address this in our overall response.
>
> **Initial bit overhead:** The average initial bit cost per TU dataset in Table 3 is 0.01 bits per edge for both ER and PU, demonstrating good amortization. We added this result to section 5.
>
>
> **Clarification of the discount factors for Table 3:** We added a column to report the discounts for all datasets in Table 3. As in Table 2, the discounts are very significant for compression performance.
>
> ## Questions
> **Extending the framework with approximate isomorphism solutions:** The tradeoff between rate and performance of the approximate version of shuffle coding is promising: Preliminary experiments show that on SZIP graphs, we can achieve speedups of multiple orders of magnitude for a few-percent increase in compression rate compared to optimal-rate shuffle coding, i. e. a 1000x speedup for a 4% rate increase on the “geom” dataset. We implemented a version of shuffle coding that has pseudo-linear runtime and completes on graphs with more than 10 million edges. The approximate method saves the “easy-to-find” redundant bits, leaving the “hard” bits out that cause most of the computational complexity when calculating the canonical ordering in nauty and traces. On all datasets we tested, there seems to be a favorable distribution between these, i. e. many easy bits vs. few hard bits. Characterizing and predicting this tradeoff would be an interesting avenue for future work.
>
> **Contribution as a general solution seems somewhat exaggerated:** We updated section 2.1.1 to clarify that our method critically depends on the availability of function to retrieve a canonical ordering (as well as a set of generators of the automorphism group) for elements of the permutable class. As mentioned in Anders and Schweitzer (2021), embedding into vertex-colored graphs, and then running nauty/traces, is the standard technique for computing the automorphism group (and canonization) of other objects. It would be great though to prove that this is always possible, and to have a general embedding method. We actually used an embedding of edge-colored graphs into vertex-colored graphs in order to canonize and compress graphs with edge attributes (which are not directly supported by nauty/traces) for our experiments. We have updated the paper to mention that we did this.
>
> ## Suggestions
> We agree with all of the reviewer’s suggestions and have implemented them in our updated paper version. This includes a simplified and better-framed definition 2.5, an updated statement regarding the initial bit overhead of rANS, and a missing clarification regarding units (all results are indeed in bits, this information was missing for Table 5).
>
> **Invertibility of encode function.** The encode function for shuffle coding *is* invertible because it takes an unordered object of type $X=\tilde{F}$ (not $F$), so the codec signature is $M×\tilde{F}→M$. We updated section 3.1 to mention this explicitly. We merely use an ordered object with arbitrary order to represent the unordered object in our implementation, as described in section 3.1. Their order has no meaning in the sense that equality of unordered objects is implemented by comparing canonized versions of their ordered representations. In practice, we use a wrapper data type called `Unordered` to define equality in this way (i. e. `Unordered(‘abc’) == Unordered(‘bca’)`), so that the universal invertibility property `c.decode(c.encode(m, x)) == (m, x)` of codecs is upheld. This is omitted in the listing of section 3.1 for simplicity.

---

> > ### Comment · Reviewer_aWcE · 2023-11-22
> > **Response to rebuttal**
> >
> > Thank you for your rebuttal.
> > I am happy to hear you found my review thoughtful, thorough and constructive, and that some of the suggestions I made were useful for improving the paper itself.
> > Your rebuttal has generally addressed most points of concern that I have.
> > I also think the preliminary exploration on extending this approach with approximate graph isomorphisms is a very interesting one, and that if the speed-up to rate trade-offs you are claiming are in fact consistent / robust across different graphs, this would make for a very compelling method.
> > I am currently erring on maintaining my score of 6 (on the grounds of the current version of the method being potentially very slow).
> > However, I think the possible improvements that could come from this extension could make this a very effective and practical compression scheme, and that __this is a potentially important factor in favour of this method that the AC could consider when assessing the paper__.

---

### Official Review · Reviewer_gbs3 · 2023-11-01

**Soundness:** 3 good
**Presentation:** 2 fair
**Contribution:** 2 fair
**Rating:** 5
**Confidence:** 2

**Summary:**

This paper focuses on the problem of compressing unordered objects. It presents a general approach called “shuffle coding” to compress different data structures with bits-back coding. After introducing the background in Section2, including data structures and the problem definition of compressing unordered objects, the authors derive Lemma3.2 and then provide the pseudo algorithm for the proposed method. The key idea is to decode an ordering as part of an encode function. The experimental results demonstrate that the proposed shuffle coding could compress different data structures with better lossless compression performance, compared with ordered ER.

**Strengths:**

It seems to be meaningful to reduce compression cost by removing the order information in data structure. The proposed shuffle coding can get a discount in lossless compression of such data structures, as illustrated by Equation 14.

**Weaknesses:**

My major concern is about the significance of the problem studied in this paper: considering the complexity, will the proposed method have wide/potential applications in practice? For my side, it seems slightly intuitive to remove the order information so that we can reduce the coding cost when we compressing graph data. Is bits-back coding necessary in this scheme? These my concern may partially be attributed to my lack of expertise in the field of compressing graphs. In addition,  Appendix C describes the modifications compared with Daniel et al., 2023b, which is also an important baseline in this paper. But for a reader that is not very familiar with the area, appendix C may be hard to understand.

**Questions:**

As I am not entirely familiar with this field, the authors are welcome to direct my attention and address my concerns with more details provided. And I’ll gladly consider increasing my initial rating.

---

> ### Author Response · Authors · 2023-11-15
>
> We thank reviewer gbs3 for the time to review our paper.
>
> We addressed the concern regarding runtime complexity in our overall response.
>
> It is intractable to directly compress unordered objects without bits-back at the optimal rate. The role of bits-back coding is to enable us to convert the problem of compressing unordered objects into a problem of compressing ordered objects, where we can apply common sequential methods such as the ones shown in Listing 1 in the paper.
>
> Appendix C describes details of an ordered graph model that we plug into shuffle coding for our experiments. It is not a baseline for shuffle coding, since we are concerned with unordered graph compression. It is not core to our method either since shuffle coding can be combined with any ordered graph codec. We merely aimed to demonstrate that a few-parameter ordered model/codec can lead to competitive results in combination with shuffle coding, and Severo et al. 2023b is a good fit for that purpose. The minor modifications lead to rate improvements as shown in Appendix F, but are not core to our method.

---

> > ### Comment · Reviewer_gbs3 · 2023-11-23
> >
> > Thanks for the feedback and confirmation on the concern regarding the limited practical value. I would like to keep the score.

---

### Author Response · Authors · 2023-11-15

We are delighted to read that all reviewers agree on the good soundness of our paper, and mostly agree that it represents a “good” contribution. Multiple reviewers highlighted the novelty, elegance and strong compression performance of the approach.

A concern regarding the practical value of our method due to its large runtime complexity was voiced multiple times. We clearly acknowledge this as a limitation in the paper. We show that shuffle coding is tractable for many practical datasets, even with the basic version presented in this paper. We leave a description of a version of shuffle coding with pseudo-linear runtime and similar rates, with wider potential for applications, for future work.

We released an updated version of the paper, following reviewers’ feedback. Most notably, the framing and content of definition 2.5 were revised for clarity, additional experimental results were added, and we now report compression speeds in Table 4.

---

### Meta-Review · Area_Chair_Qi18 · 2023-12-09

**Metareview:**

A new method for compression is valuable to the community. This paper provides a reasonable methodology. Almost all the reviewers are positive about the contribution. I recommend acceptance.

**Justification For Why Not Higher Score:**

The reviewers have all raised the issue of practical implementation of entropy coding. That makes the impact somewhat lower. Also compression is a great application area, but the methodology does not belong to the core of machine learning.

**Justification For Why Not Lower Score:**

It is a methodology paper, with deficiencies, that can be addressed in follow up works.

---

### Decision · Program_Chairs · 2024-01-16

Accept (poster)